# Leveraging VLMs for MUDA: A Category-Specific Prompting with Multi-Modal Low-Rank Adapter

## Abstract

Multi-Source Domain Adaptation (MSDA) aims to adaptively apply knowledge from multiple source pre-trained models to an unlabeled target domain. Current MSDA methods typically require extensive parameter tuning for each source model, which becomes computationally expensive, especially when dealing with numerous source domains or larger source models. With the recent advancements of Vision-Language Models (VLMs) as natural source models, the challenges of cross-domain tasks based on multi-source domains have evolved: 1) VLMs rapidly adapt to downstream tasks through prompt tuning, yet learnable prompt tokens are prone to overfitting due to limited training samples; 2) Rapidly leveraging knowledge from multiple source domains and encouraging the learning of invariant representations across these domains is a central issue; 3) The presence of visual and textual domain gaps, as well as cross-modal misalignment, can significantly impact model performance. In this paper, we propose a fine-tuning framework that integrates prompts with multimodal Low-Rank Adaptation (LoRA). This framework employs learnable prompt features as shared characteristics across different domains and utilizes multimodal LoRA matrices to represent domain-specific features for individual fine-tuning of VLMs across multiple source domains. Furthermore, it encourages interaction between fine-tuning parameters from different domains and modalities to enhance consistency. We combine all source domain-specific LoRA modules into an integrated module using a set of coefficients and adapt this integrated module to learn on the target domain. Extensive experiments demonstrate that our approach achieves significant improvements on standard image classification benchmark datasets, highlighting its effectiveness in multi-source domain adaptation tasks.

## 1 Introduction

With the rapid advancement of deep learning technologies, Multi-source Unsupervised Domain Adaptation (MUDA) has emerged as a significant research direction in the field of computer vision. The goal of MUDA is to transfer knowledge from multiple labeled source domains to an unlabeled target domain, thereby enhancing the model's adaptability capabilities on the target domain (Zhu et al., 2019a). Unlike single-source unsupervised domain adaptation, where all training samples come from a single domain, MUDA aligns more closely with real-world application scenarios where labeled images originate from multiple distinct domains. In the era of large models, existing MUDA methods that rely on comprehensive parameter tuning for each source model not only increase computational costs but also limit their feasibility in large-scale (Li et al., 2024).

In recent years, Vision-Language Models (VLMs) have achieved significant progress in multimodal learning tasks, particularly in tasks that involve understanding and generating information that combines visual and linguistic data. VLMs have propelled the development of cross-modal learning by fusing image and text information. However, as VLMs are increasingly applied, the challenges associated with MUDA continue to evolve. Firstly, VLMs can rapidly adapt to various downstream tasks through prompt tuning, with many researchers proposing fine-tuning methods based on this approach. (Zhou et al., 2022a;b; Ge et al., 2022; Bai et al., 2024; Chen et al., 2024)Yet, the reliance on learnable prompt tokens can lead to overfitting when training samples are limited, severely impact-

ing the model's adaptability and generalization capabilities (Li et al., 2023). Efficiently harnessing knowledge from multiple source domains to foster learning of cross-domain invariant representations remains a central research issue. Effective cross-domain representations can reduce biases between different source domains and improve model performance on the target domain. Moreover, the differences between visual and textual modalities and cross-modal misalignments significantly affect the overall model performance (khattak et al., 2023; Yang et al., 2024a). Specifically, visual and linguistic information exhibit substantial differences in expression forms, semantic associations, and data distributions, making models prone to information loss and comprehension barriers during cross-modal learning. Such modality misalignments not only affect the model's feature extraction capabilities but also the accuracy of final classification or regression tasks.

To address these challenges and resolve the aforementioned issues, this paper introduces a novel fine-tuning framework that integrates prompt tuning with multimodal Low-Rank Adaptation (LoRA) technology. Specifically, our approach constructs shared features across different domains through learnable prompt token features, which not only enhances the efficiency of knowledge transfer between source domains but also strengthens the model's adaptability to the target domain. Concurrently, by employing multimodal LoRA matrices, we can perform personalized fine-tuning of domain-specific features, enabling VLMs to seamlessly transition between feature spaces of different source domains. Furthermore, we introduce a new mechanism to encourage interaction between fine-tuning parameters from different domains and modalities. This interaction mechanism allows the model to learn more consistent and robust representations, thereby exhibiting stronger robustness and adaptability when handling complex multi-domain tasks. Ultimately, by combining an optimized set of coefficients, we integrate all available LoRA modules into an ensemble module for adaptive learning on the target domain. In summary, our main contributions are:

1)We propose a novel fine-tuning framework that integrates prompt tuning with multimodal Low-Rank Adaptation (LoRA) for adapting VLMs to MUDA task.
2)We introduce learnable, class-specific prompts to represent domain-invariant features and employ separate multimodal LoRA adapters for each domain to effectively capture domain-specific characteristics.
3)Furthermore, we incorporate interactions between cross-domain and cross-modal fine-tuning parameters to enhance consistency and accuracy during the learning process.

## 2 RELATED WORK

### 2.1 MULTI-SOURCE UNSUPERVISED DOMAIN ADAPTATION(MUDA)

The advent of MUDA has been prompted by the limitations of single-source unsupervised domain adaptation, which struggles to capture the diversity and complexity of real-world application scenarios. MUDA is inherently more challenging yet aligns more closely with practical needs, offering greater value. MUDA originated from the pioneering work of Yang et al. Yang et al. (2007), and has since evolved with a foundation of theoretical underpinnings (Ben-David et al., 2010; Mansour et al., 2008) and a multitude of practical applications (Duan et al., 2012; Xu et al., 2018a). Over the years, a variety of methods have been explored for MUDA tasks. Some of these methods aim to minimize the discrepancies between multiple source domains and the target domain, striving to align their distributions and mitigate the impact of domain shift, thereby reducing the target domain generalization error. For instance, MFSAN (Zhu et al., 2019a) employs Maximum Mean Discrepancy (MMD) to align distributions, while M3SDA (Peng et al., 2019a) calculates the moment distances between domains and designs a moment matching network to align feature distributions. Li et al. Li et al. (2018) utilize Wasserstein distance to diminish domain disparities. Mansour et al. Mansour et al. (2008) propose that the ideal target hypothesis can be represented as a weighted combination of the distributions of source hypotheses. Consequently, some methods aggregate predictions from different source domains to derive the final prediction for the target domain. MFSAN (Zhu et al., 2019a), after training the model with a series of regularization terms, averages the predictions from each source domain. M3SDA (Peng et al., 2019a) weights its predictions based on the accuracy derived from each source domain, while DCTN (Xu et al., 2018a) trains a domain discriminator, using its output as the basis for weighting across multiple source domains. Wu et al. Wu et al. (2020) calculate the conditional Wasserstein distance from different source domains to the target domain and use these results for weighting. Beyond the aforementioned approaches, other methods have been

devised to address MUDA challenges. MOST (Nguyen et al., 2021a) leverages optimal transport theory and employs imitation learning between student and teacher classifiers to enhance MUDA performance. STEM (Nguyen et al., 2021b) incorporates adversarial learning concepts, mapping multiple source teacher networks and a target domain student network to the same feature space, enabling the student classifier to effectively mimic the teachers for classification predictions. PFSA (Fu et al., 2021a) selectively aligns features from multiple source domains that are compatible with the target domain by minimizing the loss of intra-class sample clustering and maximizing inter-class distance differences. MPA (Chen et al., 2024) employs CLIP as the backbone and designs prompts that encompass both domain-invariant and domain-specific features, training such prompts for each source and target domain pair. It uses a contrastive loss to train and bridge the gap between source and target domains.

## 2.2 VISION LANGUAGE MODELS(VLMS)

Vision-Language Models (VLMs) integrate natural images with linguistic supervision, diverging from traditional vision models that rely on discrete labels for category prediction. Instead, VLMs harness supervision from natural language, thereby leveraging the semantics encapsulated in text to a greater extent. A quintessential example is the CLIP model (Radford et al., 2021b), which employs a contrastive learning approach in the feature representation space. It aggregates positive pairs—related images and texts—while separating negative pairs—unrelated instances—to establish correspondences between text and images. This approach fosters a deeper understanding of the complex interplay between text and images. The multimodal representations of vision and language also render their applications more flexible and diverse.Recently, VLMs like CLIP (Radford et al., 2021b), ALIGN (Jia et al., 2021), FILIP (Yao et al., 2022), LiT (Zhai et al., 2022), and Florence (Yuan et al., 2021) have been utilized as foundational models for a broad range of downstream tasks. These models are typically trained on web-scale datasets; for instance, CLIP and ALIGN are trained on 400 million and 1 billion image-text pairs, respectively, to develop their multimodal networks. Although these pre-trained VLMs learn generalized representations, effectively adapting them to downstream tasks remains a challenging issue.There has been significant work on devising methods to adapt VLMs to downstream tasks such as few-shot image recognition (Kim et al., 2022; Zhang et al., 2021), image segmentation (Ding et al., 2022; Li et al., 2022; Lüddecke & Ecker, 2022; Rao et al., 2022), and object detection (Feng et al., 2022; Maaz et al., 2022; Bangalath et al., 2022; Zhou et al., 2022c). Given VLMs' robust capability to integrate semantic information, they are particularly well-suited for unsupervised domain adaptation. In this work, we also employ the CLIP model as our foundational model and design methods to better adapt it to MUDA tasks, thereby enhancing performance.

## 2.3 FINE-TUNING METHODS BASED ON PRE-TRAINED MODELS

Pre-trained models have garnered extensive knowledge and generalized representations, yet efficiently adapting them to downstream tasks remains a formidable challenge. The most straightforward adaptation approach is full fine-tuning in the downstream task, but this becomes increasingly expensive and even infeasible as model parameter counts grow. Consequently, researchers have delved into more efficient fine-tuning techniques. Prompt tuning stands out as an efficient parameter tuning method that guides the model to achieve desired outputs by designing learnable input queries with minimal parameters while keeping the original model frozen. CoOp (Zhou et al., 2022a) pioneered the introduction of soft prompts in VLMs, demonstrating that appropriate textual prompts could enhance image recognition performance. Building on CoOp, CoCoOp (Zhou et al., 2022b) further addressed overfitting issues by learning a lightweight neural network to dynamically generate prompts for each input image. DAPrompt (Ge et al., 2022) embedded domain-specific knowledge into prompts to facilitate domain alignment. VPT (Jia et al., 2022) successfully applied prompts to the image encoder, yielding promising results. DoPrompt (Zheng et al., 2022) leveraged prompt learning to embed source domain knowledge into domain prompts for predicting target domain outcomes. MaPLe (khattak et al., 2023) introduced the novel concept of multimodal prompt learning, simultaneously learning prompts on both visual and textual branches to improve representation consistency. PDA (Bai et al., 2024) also employed multimodal prompts and designed a dual-branch prompt tuning paradigm, achieving impressive results. Adapter tuning is another popular efficient parameter tuning method that inserts additional trainable parameters into pre-trained models to adapt to downstream tasks. A variety of adapters such as Clip-Adapter (Gao et al., 2021),

Tip-Adapter (Zhang et al., 2021), VL-Adapter (Yi-Lin Sung, 2022), SVL-Adapter (Pantazis et al., 2022), crossmodal-adapter (Jiang et al., 2022), MV-Adapter (Valipour et al., 2023), and MMA (Yang et al., 2024a) have been integrated into backbone networks to accomplish a wide array of downstream tasks. Some adapters, like Clip-Adapter (Gao et al., 2021) and Tip-Adapter (Zhang et al., 2021), are added to a single modality branch, while some adapters, such as crossmodal-adapter (Jiang et al., 2022) and MMA (Yang et al., 2024a), facilitate cross-modal interactions.

# 3 METHOD

In Multi-source Unsupervised Domain Adaptation (MUDA), there are M distinct underlying source distributions, denoted as $\left\{p_{s_j}(x,y)\right\}_{j=1}^{M}$. Labeled source domain data $\left\{\left(X_{s_j}, Y_{s_j}\right)\right\}_{j=1}^{M}$ are drawn from these distributions, where $X_{s_j} = \left\{x_i^{sj}\right\}_{i=1}^{N_{sj}}$ represents the samples from source domain $j$, and $Y_{s_j} = \left\{y_i^{sj}\right\}_{i=1}^{N_{sj}}$ corresponds to the true labels of the samples, with $N_{sj}$ indicating the dataset size of source domain $j$. The distribution of the target domain is $p_t(x,y)$, and for all $\forall j \in \{1, 2, \cdots, M\}$, it holds that $p_t(x,y) \neq p_{s_j}(x,y)$. From this, we sample the target domain data $X_t = \{x_i^t\}_{i=1}^{N_t}$, where $N_t$ denotes the dataset size of the target domain, but without the corresponding true labels $Y_t$. All source and target domains share the same K classes. Below, we will first provide a brief overview of the pre-trained CLIP architecture, followed by our proposed method to address the MUDA problem.

## 3.1 REVIEW OF CLIP

Our approach is designed and constructed upon the pre-trained Visual-Linguistic Model CLIP, which comprises a language branch and a visual branch. These branches utilize transformers and Vision Transformers (ViT)(Dosovitskiy et al., 2021b) to encode corresponding text descriptions and images $I \in \mathbb{R}^{H \times W \times 3}$. The specific processes are detailed as follows:

Language Branch: For input text, it first undergoes tokenization, which is the process of segmenting text into tokens. Subsequently, these tokens are mapped to 512-dimensional embedding vectors, resulting in $W_0 = \left[w_0^1, w_0^2, \cdots, w_0^N\right] \in \mathbb{R}^{N \times 512}$ which are then passed to an encoder with K Transformer layers. $W_i$ is transformed through the corresponding transformer layer $(\mathcal{L}_i)$ in the language branch to obtain the next stage representation:

$$[W_{i+1}] = \mathcal{L}_i(W_i) \qquad i = 0,\ 1, \cdots, K-1 \tag{1}$$

After passing through all transformer layers, $W_0$ yields the final text embedding representation $W_k$. Through TextProj, the text embedding corresponding to the last token of $W_k$ is projected into the common V-L latent embedding space to obtain the final text representation z.

$$z = TextProj\left(w_K^N\right) \quad z \in \mathbb{R}^{512} \tag{2}$$

Visual Branch: For input images, they are first divided into patches of size M, and then these patches are projected into 768-dimensional patch embeddings, i.e., $E_0 \in \mathbb{R}^{M \times 768}$. The patch embeddings $E_i$ and the learnable class (CLS) token $c_i$ are fed into a ViT with K Transformer layers for encoding. They are transformed through the corresponding Transformer layer $(V_i)$ in the visual branch to obtain the next stage representation,:

$$[c_{i+1}, E_{i+1}] = V_i([c_i, E_i]) \quad i = 0,\ 1, \cdots, K-1 \tag{3}$$

After passing through all transformer layers, the class (CLS) token $c_i$ results in $c_K$, which is projected through ImageProj into the common V-L latent embedding space to obtain the final image representation $x$:

$$x = ImageProj(c_K) \quad x \in \mathbb{R}^{512} \tag{4}$$

Prediction and Inference: For image classification tasks, the text input consists of manually designed prompts based on class labels $y \in \{1, 2, \cdots, C\}$(e.g., 'a photo of a ¡category¿'), where ¡category¿ is replaced by the class label, resulting in a set of C prompts. During prediction, while the prompts are input into the language branch, the image to be predicted is input into the visual branch. Then, in the common V-L latent embedding space, the cosine similarity $(sim(\cdot))$ between the two is calculated

using a temperature parameter $\tau$, and the class of the prompt with the maximum value is taken as the final predicted label for the image.

$$p\left(\hat{y}\mid x\right) = \frac{\exp\left(sim\left(x, z_{\hat{y}}\right)/\tau\right)}{\sum\limits_{i=1}^{C} exp\left(sim\left(x, z_i\right)\right)} \tag{5}$$

## 3.2 LEVERAGING VLMs FOR MUDA

To better adapt the CLIP model for downstream MUDA task, we have designed a framework that employs class-specific prompts and multimodal Low-Rank Adaptation (LoRA) adapters, as illustrated in Fig .1. Our goal is to utilize class-specific prompt features as shared characteristics across different domains, while employing multimodal LoRA matrices to represent domain-specific features for fine-tuning the CLIP model.

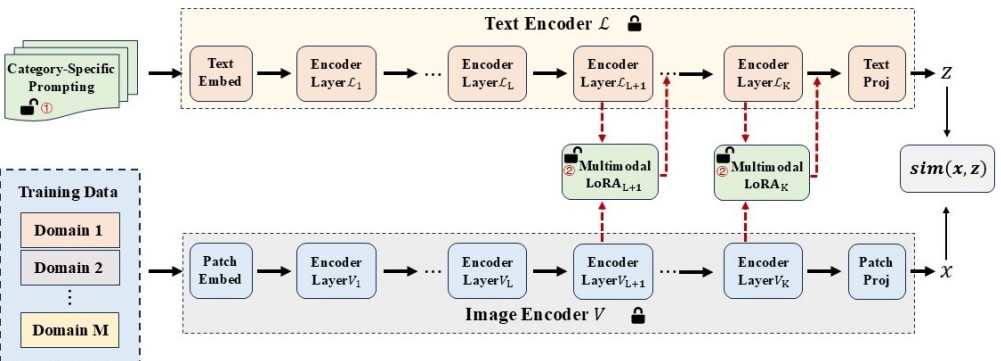

Figure 1: The overall architecture diagram of our method

### 3.2.1 CATEGORY-SPECIFIC PROMPTING

Designing learnable prompts to replace handcrafted ones allows for more nuanced and flexible fine-tuning of the model, leading to promising results. MPA[Multi-prompt alignment for multi-source unsupervised domain adaptation](Chen et al., 2024) method trains distinct prompts for different domains to accomplish MUDA tasks and has achieved certain effects. However, Li et al. Li et al. (2023) argue that learnable prompts trained on limited samples are prone to overfitting, which may undermine the adaptability and generalization capabilities of the pre-trained model. Therefore, training domain-specific prompts for each source domain may not be an optimal choice. Moreover, considering that in MUDA tasks, all source domains and the target domain share the same label space, i.e., identical classes, it is beneficial to use data from all source domains to jointly train a set of learnable class-specific prompts. This approach not only leverages precise and flexible prompts for model fine-tuning but also reduces the risk of overfitting due to limited training samples. Specifically, we introduce bb learnable tokens $\left\{P^i \in \mathbb{R}^{512}\right\}_{i=1}^{b}$ into the language branch of CLIP and merge them with class labels, with the prompt format as follows:

$$\left[P^1\right]\left[P^2\right]\ldots\left[P^b\right]\left[CLASS\right] \tag{6}$$

Where, [CLASS]$[CLASS]$ is replaced according to the label. The reason for fixing the label information at the last token of the prompt, rather than at the beginning or in the middle, is that after the prompt passes through all the transformer layers of the CLIP model's language branch, the text embedding corresponding to the last token is projected into the common V-L latent embedding space. Therefore, fixing the label information at the end of the prompt allows for more effective utilization of its semantic content.

### 3.2.2 MULTI-MODAL LOW-RANK ADAPTER

Fine-tuning the CLIP model solely with prompts is insufficient for effectively addressing the complexities of Multi-source Unsupervised Domain Adaptation (MUDA) tasks. Therefore, we introduce adapters to enable the model to learn domain-specific knowledge through adapter parameters. Adapting CLIP's original model structure directly through adapters would require a substantial number of parameters. To address this, we incorporate Low-Rank Adaptation (LoRA) (Hu et al., 2022) matrices.

The LoRA matrix operates based on the concept of the "intrinsic rank" required for downstream tasks, modeling the incremental update of pre-trained weights as the product of two small matrices, A and B. For an input x, a hidden state h, and a weight matrix $w \in \mathbb{R}^{d_1 \times d_2}$, the modified forward pass after applying the LoRA module is:

$$h = Wx + \gamma \Delta Wx = Wx + \gamma BAx \tag{7}$$

Where, $A \in \mathbb{R}^{r \times d_2}$, $B \in \mathbb{R}^{d_1 \times r}$ , and $\Delta W \in \mathbb{R}^{d_1 \times d_2}$ is of rank r, with r typically much smaller than $\{d_1, d_2\}$, and $\gamma$ is a scaling factor. Matrix A is randomly initialized using Kaiming initialization, while B is filled with zeros. This implies no incremental update before training, thus the output remains unchanged. We apply the low-rank matrix to the attention matrices of the transformer structure, which typically consists of LL stacked blocks, each containing a Multi-Head Attention (MHA) (Scao et al., 2022) module:

$$had_i = Softmax\left(\frac{xW_{q_i}\left(xW_{k_i}\right)^T}{\sqrt{d}}\right)(xW_{v_i}) \tag{8}$$

$$MHA(x) = concat(had_1, \cdots, had_H) W_o \tag{9}$$

where d is a scaling factor, and $W_{K_i}$, $W_{Q_i}$, $W_{V_i}$, $W_o$ are weight matrices corresponding to key, query, value, and output matrices, respectively.

Since CLIP's visual and language branches both contain transformer structures, we introduce corresponding LoRA matrices in both branches. Based on the experiments and observations of Yang et al. (2024b), higher layers in pre-trained text and image encoders contain discriminative dataset-specific representations, while lower layers contain more generalizable representations across different datasets. We aim to utilize the parameters of the LoRA matrices to learn domain-specific features, making it suitable to introduce LoRA in higher transformer layers. Formally, for the text encoder L, we add an adapter $Ad^t$ from the L-th transformer block and modify the equation1as follows:

$$[W_{i+1}] = \mathcal{L}_i (W_i) \qquad i = 0, 1, \cdots, L - 1 \tag{10}$$

$$[W_{j+1}] = \mathcal{L}_j (W_j) + \alpha Ad_j^t (W_j) \qquad j = L, L + 1, \cdots, K - 1 \tag{11}$$

where $\alpha$ is a balance coefficient between adapter knowledge and pre-trained knowledge. When $\alpha$=0, the model uses the original transformer blocks without adapter parameters. Similarly, we introduce an adapter $Ad^v$ for the visual encoder $V$, starting from the L-th transformer block:

$$[c_{i+1}, E_{i+1}] = V_i ([c_i, E_i]) \qquad i = 0, 1, \cdots, L - 1 \tag{12}$$

$$[c_{j+1}, E_{j+1}] = V_j ([c_j, E_j]) + \alpha Ad_j^v ([c_j, E_j]) \qquad j = L, L + 1, \cdots, K - 1 \tag{13}$$

If we only add independent LoRA matrices to the two branches, there would be no interaction or influence between the two modalities, lacking collaborative synergy. The significant semantic gap between branches would make it difficult for the model to align. Therefore, connecting the two branches is crucial. We propose using a shared projection layer $W_{Lshare}$ to aggregate features from different modalities. Before aggregation, a projection layer $W_{Lout}$ is needed to transform the features from the low-rank matrices of different branches to the same dimension. After aggregation, a projection layer $W_{Lin}$ is used to transform the feature dimensions back to match the dimensions of the low-rank matrices of each branch. This process can be summarized as follows, for the text encoder:

$$\Delta W = B_L^t A_L^t \tag{14}$$

$$Ad_L^t(\Delta W) = W_{Lin}^t \cdot \delta \left( W_{Lshare} \cdot \delta \left( W_{Lout}^t \cdot \Delta W \right) \right) \tag{15}$$

where $B_L^t$ and $A_L^t$ represent the low-rank matrices B and A added in the L-th transformer layer of the text encoder. Similarly, for the visual encoder:

$$\Delta W = B_L^v A_L^v \tag{16}$$

$$Ad_L^v(\Delta W) = W_{Lin}^v \cdot \delta \left( W_{Lshare} \cdot \delta \left( W_{Lout}^v \cdot \Delta W \right) \right) \tag{17}$$

where $B_L^v$ and $A_L^v$ represent the low-rank matrices B and A added in the L-th transformer layer of the visual encoder, and $W_{Lshare}$ represents the shared projection layer corresponding to the L-th transformer layer shared by the two branches. It is important to emphasize that the shared projection acts as a bridge between the two modalities, allowing gradients to propagate between them, thus better aligning different modality features. The specific structure is shown in the Fig .2

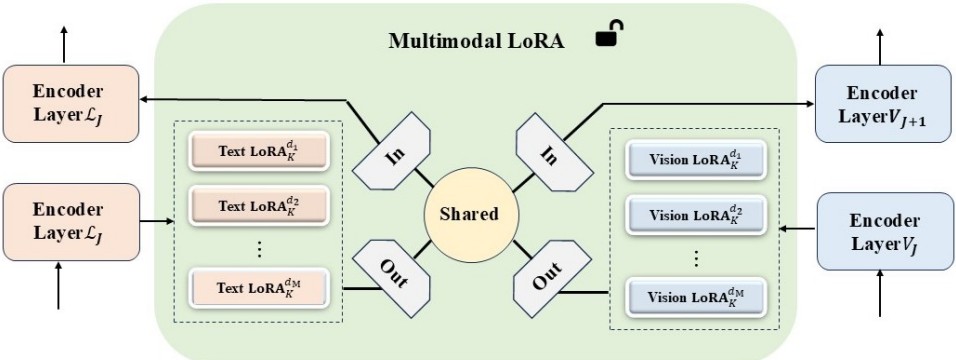

Figure 2: Multimodal LoRA Matrix and Interactive Structure

### 3.2.3 TRAINING STRATEGY AND INFERENCE PROCESS

Our proposed method incorporates learnable class-specific prompts and multi-modal LoRA matrices. Training all fine-tuning parameters simultaneously may lead to suboptimal performance due to the different meanings and representations of various parameters, which could interfere with each other when trained together. Therefore, a strategic approach is needed to train different fine-tuning parameters.

We utilize class-specific prompts to represent knowledge shared across all domains and domain-specific multi-modal LoRA matrices to represent knowledge unique to each domain. The training process consists of two steps. In the first step, we train the class-specific prompts using all data, including data from all source domains and the target domain, without updating the LoRA matrix parameters. In the second step, we freeze the learned prompts and only update the parameters related to the LoRA adapters. During training, we minimize the cross-entropy loss given the true labels $y_i^s$ for each source domain, as shown in Equation18:

$$\mathcal{L}_s = -\frac{1}{N_s} \sum_{i=1}^{N_s} logP\left( \hat{y}_i^s = y_i^s \right) \tag{18}$$

where $N_s$ denotes the total number of samples in the source domain, and $\hat{y}_i^s$ is represents the model's predicted outcome. To further leverage unlabeled data, we generate pseudo-labels for the target domain. We select the class with the highest predicted probability from K classes as the pseudo-label $y^t$ for the training data $x^t$:

$$y^t = \underset{k}{argmax} P\left( \hat{y}^t = k \middle| x^t \right), \qquad k = \{1, 2, \cdots K\} \tag{19}$$

We only generate pseudo-labels for unlabeled data where the maximum predicted probability exceeds a fixed threshold $\tau_{label}$ of pseudo-label quality:

$$\mathcal{L}_u = -\frac{1}{N_t} \sum_{i=1}^{N_t} \mathbb{I}\left\{ P\left( \hat{y}_i^t = y_i^t \middle| x_i^t \right) \geq \tau_{label} \right\} logP\left( \hat{y}_i^t = y_i^t \middle| x_i^t \right) \tag{20}$$

where $\mathbb{I}\{\cdot\}$ is the indicator function. Overall, our method is trained end-to-end, and the total loss is:

$$\mathcal{L} = \mathcal{L}_s\left(\chi^{s_i}\right) + L_t\left(\chi^t\right) \quad i = \{1, 2, \cdots M\} \tag{21}$$

where $M$ is the total number of source domains. The entire training process can be found in Algorithm 1 of the appendix. During inference, we incorporate the fine-tuned class-specific prompts into the original pre-trained model and amalgamate the multimodal LoRA matrix modules that were trained across different domains. The integrated LoRA parameters are then incorporated back into the model for making predictions.

## 4 EXPERIMENT

### 4.1 EXPERIMENTAL SETUP

**Datasets**: We conducted extensive experiments on three popular MUDA datasets to evaluate our model: (1) **Office-31** (Saenko et al., 2010): This small-scale dataset comprises images from Amazon (A), Webcam (W), and DSLR (D), totaling 4,110 images across 31 categories. We utilized all domain combinations to construct three MUDA tasks: A,W→D;A,D→W; W,D→A. (2) **Office-Home** (Venkateswara et al., 2017): A medium-scale dataset consisting of 15,588 images spanning 65 categories across four distinct domains: Artistic images (A), Clip Art (C), Product images (P), and Product images (R). We employed all domain combinations to formulate four MUDA tasks: C,P,R→A;A,P,R→C; A,C,R→P; A,C,P→R. (3) **DomainNet** (Peng et al., 2019a): A large-scale dataset comprising approximately 600,000 images belonging to 345 categories across six different domains: Clipart, Infograph, Painting, Quickdraw, Real, and Sketch. Using all domain combinations, we established six MUDA tasks→Clp;→Inf;→Pnt;→Qdr;→Rel;→Skt.

**Implementation Details.**: We employed CLIP as our base model, utilizing the Vision Transformer (ViT-B/16) (Dosovitskiy et al., 2021a)for the image encoder. We loaded the pre-trained weights, which were kept frozen throughout the experimental process. The prompts were trained for 25 epochs using a mini-batch Stochastic Gradient Descent (SGD) optimizer with a batch size of 32. The initial learning rate was set to 0.003, and we applied a cosine annealing schedule for learning rate decay (Loshchilov & Hutter, 2016). Low-rank matrices were applied exclusively to the query, key, and value matrices with a rank r=2. We regularized the input to the LoRA modules through a dropout layer with a probability p=0.25 (Feng et al., 2022). Regarding hyperparameters, the length of the class-specific prompt tokens was set to 16. The pseudo-labeling threshold  was set to 0.5 to ensure the generation of reliable labels. **Compared Methods and Evaluation Metric.**:To validate the effectiveness of our model, we have selected the following MUDA models as our multi-source baselines:DAN(Long et al., 2015), D-CORAL(Sun & Saenko, 2016),DCTN(Xu et al., 2018b), MDDA(Zhao et al., 2019),MFSAN(Zhu et al., 2019b),MPA  (Chen et al., 2024), and so on. These models represent a cross-section of state-of-the-art approaches in multi-source domain adaptation, providing a robust framework for comparative analysis.Our evaluation metrics will be aligned with standard practices in the field, focusing on accuracy and robustness across various domain shifts.

### 4.2 OVERALL COMPARISON

**Results on the Office-31 dataset.**As shown in Table 1, our model achieved an average accuracy of 85.7%, outperforming current MUDA methods. Specifically, across three distinct MUDA tasks, our performance saw a noticeable improvement in each case.

**Results on the Office-Home dataset.**According to the results presented in Table 2, our approach achieved an average accuracy of 77.7%, marking a 2.3% overall enhancement compared to the previous MPA method. The computational outcomes indicate that the model's accuracy when adapting to domains A and C is relatively lower, primarily due to the substantial gap between these domains.

**Results on the DomainNet dataset.**The results in Table 3 demonstrate that our method achieved an average accuracy of 54.8%, which represents an improvement over previous methods. We consider this dataset to be particularly challenging for several reasons: Firstly, DomainNet includes images across 345 categories within each domain, which is more than any other MSDA dataset. The sheer number of categories greatly increases the difficulty of fine-tuning the model with class-specific prompts. Secondly, there is a pronounced distribution shift between different domains, especially

Table 1: Offce-31 Dataset Performance Comparison

| **Standards** | Method | A,D →W | A,W →D | D,W →A | Avg |
|---|---|---|---|---|---|
| Source Combine | DAN(Long et al., 2015) | 96.2 | 98.8 | 54.9 | 83.3 |
| | MCD(Saito et al., 2018a) | 96.2 | 99.5 | 54.4 | 83.4 |
| Multi source | DCTN(Xu et al., 2018b) | 96.9 | 99.6 | 54.9 | 83.8 |
| | M3SDA(Peng et al., 2019a) | 96.2 | 99.4 | 55.4 | 83.7 |
| | PFSA(Fu et al., 2021b) | 97.4 | 99.7 | 57.0 | 84.7 |
| | ours | 98.0 | 99.8 | 58.0 | 85.7 |

Table 2: Office-home Performance Comparison

| | Method | C,P,R →A | A,P,R →C | A,C,R →P | A,C,P →R | AVG |
|---|---|---|---|---|---|---|
| Zero-Shot | CLIP(Radford et al., 2021a) | 71.5 | 50.2 | 81.3 | 82.4 | 71.4 |
| Source Combine | DAN(Long et al., 2015) | 68.5 | 59.4 | 79.0 | 82.5 | 72.4 |
| | DANN(Ganin et al., 2016) | 68.4 | 59.1 | 79.5 | 82.7 | 72.4 |
| | D-CORAL(Sun & Saenko, 2016) | 68.1 | 58.6 | 79.5 | 82.7 | 72.2 |
| | DAPL(Ge et al., 2023) | 72.8 | 51.9 | 82.6 | 83.7 | 72.8 |
| Multi source | MDDA(Zhao et al., 2019) | 66.7 | 62.3 | 79.5 | 79.6 | 71.0 |
| | MFSAN(Zhu et al., 2019b) | 72.1 | 62.0 | 80.3 | 81.8 | 74.1 |
| | SImpAI50(Venkat et al., 2021) | 70.8 | 56.3 | 80.2 | 81.5 | 72.2 |
| | MPA(Chen et al., 2024) | 74.8 | 54.9 | 86.2 | 85.7 | 75.4 |
| | ours | 75.0 | 62.1 | 87.0 | 86.5 | 77.7 |

between Quickdraw and the others. Such distribution variations heighten the challenge of refining informative features, as misalignments may occur when pseudo-labels are not reliable.

## 4.3 FURTHER ANALYSIS

**Selection of Pseudo-Labeling Prompts.** The extent to which our model leverages target domain data significantly impacts the ultimate prediction accuracy on that domain. Given the absence of true labels for the target domain data, it is imperative to first assign pseudo-labels to these data samples, where the quality of the pseudo-labels plays a crucial role. In this regard, we explored the use of both manually crafted and learnable prompts. Our findings indicate that manually designed prompts, specifically 'a photo of a [CLS]', outperformed learnable prompts. The initial weakness of learnable prompts when randomly initialized resulted in a limited number of target domain data

Table 3: DomainNet Performance Comparison

| | Method | →Clp | →Inf | →Pnt | →Qdr | →Rel | →Skt | AVG |
|---|---|---|---|---|---|---|---|---|
| Zero-Shot | CLIP(Radford et al., 2021a) | 61.3 | 42.0 | 56.1 | 10.3 | 79.3 | 54.1 | 50.5 |
| Source Combined | DANN(Ganin et al., 2016) | 45.5 | 13.1 | 37.0 | 13.2 | 48.9 | 31.8 | 32.6 |
| | MCD(Saito et al., 2018b) | 54.3 | 22.1 | 45.7 | 7.6 | 58.4 | 43.5 | 38.5 |
| | DAPL(Ge et al., 2023) | 62.4 | 43.8 | 59.3 | 10.6 | 81.5 | 54.6 | 52.0 |
| Multi source | M³SDA-$\beta$(Peng et al., 2019b) | 58.6 | 26.0 | 52.3 | 6.3 | 62.7 | 49.5 | 42.6 |
| | SImpAI50(Venkat et al., 2021) | 66.4 | 26.5 | 56.6 | 18.9 | 68.0 | 55.5 | 48.6 |
| | PFSA(Fu et al., 2021b) | 64.5 | 29.2 | 57.6 | 17.2 | 67.2 | 55.1 | 48.5 |
| | PTMDA(Ren et al., 2022) | 64.5 | 29.2 | 57.6 | 17.2 | 67.2 | 55.1 | 48.5 |
| | MPA(Chen et al., 2024) | 65.2 | 47.3 | 62.0 | 10.2 | 82.0 | 57.9 | 54.1 |
| | ours | 67.0 | 42.3 | 62.5 | 15.0 | 84.1 | 57.8 | 54.8 |

samples with predicted pseudo-labels surpassing the threshold $\tau_{label}$ . This limitation consequently led to a reduced application of the model on the target domain data.

**Multimodal LoRA Matrix Selection.** Initially, we explored the incorporation of LoRA matrices into individual modalities of the CLIP model by adding them to the visual encoder and textual encoder separately to assess performance. Subsequently, we introduced LoRA matrices to both the visual and textual encoders, with the matrices operating independently without interaction. We then tested the performance under this configuration. Finally, we integrated multimodal LoRA matrices and facilitated interaction between them through a shared projection layer, further testing the performance. By comparing the model performance across these various scenarios, it was evident that the use of multimodal LoRA matrices with a shared projection layer for interaction yielded the best results. This finding validates the effectiveness and necessity of our design approach. Additionally, we investigated the impact of adding multimodal LoRA matrices at different transformer layers within the encoders and found that incorporating them at higher layers of the encoders resulted in better performance.

**Hyperparameter Selection.** We conducted a thorough examination of the hyperparameters for the pseudo-labeling threshold $\tau_{label}$ and the length of the prompt tokens b. Specifically, we explored $\tau_{label}$ in the set 0.4,0.5,0.6,0.7,0.8 and b in the set 8,12,16,20. As labellabel increases, the quality of the pseudo-labels improves, but fewer images are input into the model, which can harm overall performance. Conversely, the general trend for the prompt token length is that longer prompts yield better performance. Therefore, we selected $\tau_{label} = 0.5$ and b=16 to strike a balance between performance and efficiency. When applying the LoRA matrices, a higher rank corresponds to a larger number of fine-tuning parameters, which may lead to overfitting when the training data is limited. Hence, we opted for LoRA matrices with a rank r=2.

## 5 CONCLUSION

This paper presents a novel framework leveraging Vision-Language Models (VLMs) for Multi-Source Unsupervised Domain Adaptation (MUDA). We introduce a fine-tuning approach that integrates category-specific prompting with multimodal Low-Rank Adaptation (LoRA) to enhance model adaptability across domains. Extensive experiments on benchmark datasets demonstrate our method's effectiveness in improving classification accuracy, showcasing significant advances over existing MUDA techniques. Our work provides a promising direction for adapting VLMs to diverse real-world applications.

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

# A   APPENDIX

The entire training process is shown in Algorithm 1 below

---

### Algorithm 1: Overall Training Procedure

**Input**: Pre-trained CLIP model parameters $\theta$, number of source domains M, labeled multi-source domain data $\chi_{S_1}, \chi_{S_2}, \cdots, \chi_{S_N}$ along with corresponding labels $y$, unlabeled target domain data $\chi_t$, manually designed prompts $P^{head}$, pseudo-label threshold $\tau_{label}$, learnable class-specific prompts $P^{learn}$, low-rank matrices $A_i$ and $B_i$, and number of epochs T.

**Output**: Fine-tuned target model $\theta^t$ (including the optimized prompts $P^*_{learn}$ and multi-modal LoRA parameters corresponding to different domains).

**Procedure**:

1:  **Initialization**: Set $P^{head} = $ 'a photo of a $[CLS]$.', $P^{learn}$, $A_i$, $B_i$, set $\theta^t = \theta$.
2:  Use $P^{head}$ and $\tau_{label}$ to generate pseudo-labels for the target domain data according to Equation 10, obtaining $\chi_t^{label}$.
3:  Load all source domain data $\chi_{S_1}, \chi_{S_2}, \cdots, \chi_{S_N}$ and the pseudo-labeled target domain data $\chi_t^{label}$.
4:  ======================= Step 1 =========================
5:  Freeze LoRA matrix parameters and train only the class-specific prompts:
6:  **for** t=1 to T **do**
7:      Sample a batch from $\chi_{S_1}, \chi_{S_2}, \cdots, \chi_{S_N}, \chi_t^{label}$;
8:      Forward the prompt $P^{learn}$ and the batch data through $\theta^t$
9:      Update $P^{learn}$ using Equations 11 and 12.
10:  **end for**
11:  Obtain the optimally trained class-specific prompts $P^*_{learn}$.
12:  ======================= Step 2 =========================
13:  Freeze the trained prompts and train only the LoRA parameters for each domain:
14:  **for** t=1 to T **do**
15:      Incorporate the optimally trained prompt $P^*_{learn}$ into $\theta^t$;
16:      Sample a batch from $\chi_{S_1}, \chi_{S_2}, \cdots, \chi_{S_N}, \chi_t^{label}$;
17:      Separate the batch data based on their origin domain;
18:      Forward the multi-modal LoRA parameters corresponding to each domain through the respective domain data;
19:      Update the corresponding multi-modal LoRA parameters using Equations 11 and 12.
20:  **end for**
21:  Acquire the most effective multi-modal LoRA parameters for each domain.
22:  **return** the target model $\theta^t$.

---

