# OpenReview forum: "Leveraging VLMs for MUDA：A Category-Specific Prompting with Multi-Modal Low-Rank Adapter"
_ICLR.cc/2025/Conference — ICLR 2025 Conference Withdrawn Submission_

### Official Review · Reviewer_MRmR · 2024-11-01

**Soundness:** 2
**Presentation:** 3
**Contribution:** 3
**Rating:** 5
**Confidence:** 3

**Summary:**

This paper tackles the challenge of Multi-Source Domain Adaptation (MSDA) in Vision-Language Models (VLMs), aiming to transfer knowledge from multiple source domains to an unlabeled target domain. The authors highlight issues in current MSDA methods, including computationally costly parameter tuning and the risk of overfitting with limited training samples when using learnable prompts. To address these, they introduce a fine-tuning framework that combines prompt tuning with multimodal Low-Rank Adaptation (LoRA). This framework is designed to (1) utilize shared, learnable prompt features across domains to capture invariant features; (2) employ domain-specific multimodal LoRA matrices for targeted fine-tuning across various source domains; and (3) encourage interactions between domain-specific and modality-specific parameters to enhance consistency across domains and modalities. The proposed framework consolidates all source LoRA modules into an integrated module adapted for the target domain. Their experimental results demonstrate significant performance improvements on standard image classification benchmarks, underscoring the approach’s effectiveness in MSDA tasks.

**Strengths:**

(1)Originality: This paper presents a unique approach by combining category-specific prompting and multimodal Low-Rank Adaptation (LoRA) for MSDA within VLMs. The integration of LoRA in both visual and textual modalities, coupled with category-specific prompts, addresses specific challenges in MSDA, such as overfitting with limited data and cross-modal misalignment. This approach stands out for its creativity in bridging domain-invariant and domain-specific features, which could inspire further work in efficient domain adaptation methods.
(2)Quality: The experimental results showcase a clear performance gain on popular benchmarks, especially in challenging datasets like DomainNet.
(3)Clarity: The methodology section is organized clearly, allowing readers to understand the purpose and implementation of each component within the proposed framework. The interaction mechanism between fine-tuning parameters in different domains and modalities is particularly well-justified in the context of MSDA challenges.
(4)Significance:
The proposed framework is flexible and adaptable to different datasets and domains, showcasing strong potential for other applications, where training data is scarce or where multiple distinct domains are involved.  The approach is valuable not only for the academic MSDA community but also for practical settings that benefit from efficient model fine-tuning across domains.

**Weaknesses:**

(1) Lack of Experimental Data for Overfitting Mitigation:
Although a key motivation of the paper is to address the overfitting issues in VLMs caused by limited training samples and reliance on learnable prompt tokens, the paper does not provide sufficient experimental metrics to demonstrate how effectively the proposed approach mitigates overfitting. Specific metrics, such as validation accuracy trends across training epochs, performance under varied training data sizes, or comparisons of overfitting rates between prompt tuning alone versus prompt tuning combined with LoRA, would provide clearer evidence supporting this motivation.
(2) Ablation Study on Fine-Tuning Components:
The authors have touched upon the different scenarios of Multimodal LoRA Matrix Selection; however, given the complexity of this integrated approach, a more rigorous ablation study with precise data is necessary to substantiate their claims. Specifically, a comparison of performances with and without prompt tuning, with only LoRA, or with varying ranks and configurations of LoRA matrices would provide deeper insights into the roles and benefits of each part.
(3) Minor Errors and Formatting Issues:
The paper contains several minor errors that could detract from its overall quality. For instance, in line 265, the term "[CLASS]" appears consecutively. Additionally, in line 410, the phrase "Compared Methods and Evaluation Metric" should begin a new line for improved clarity and presentation. It is recommended that the authors carefully proofread the entire manuscript to address these and other minor issues, ensuring a polished and professional presentation.

**Questions:**

please see the weakness

---

### Official Review · Reviewer_j1o4 · 2024-11-02

**Soundness:** 2
**Presentation:** 3
**Contribution:** 2
**Rating:** 3
**Confidence:** 5

**Summary:**

This paper addresses the challenges of adapting Vision-Language Models (VLMs) to multi-source domain adaptation tasks. It introduces a novel framework that incorporates class-specific prompting with multimodal Low-Rank Adaptation (LoRA) matrices. The proposed method uses learnable prompt tokens as shared features across different domains and employs domain-specific LoRA matrices for targeted fine-tuning. This approach enhances the transfer of knowledge from multiple source domains to an unlabeled target domain, improving model adaptability and performance across different domain shifts. The paper reports significant improvements in classification accuracy on benchmark datasets, demonstrating the effectiveness of the method.

**Strengths:**

1. The paper demonstrates the robustness of the proposed method through extensive experiments on standard datasets like Office-31, Office-Home, and DomainNet, which enhances the credibility of the results.
2. The paper explains the construction of the proposed model clearly, covering both the prompting mechanism and the LoRA matrix integration process.

**Weaknesses:**

1. Although the paper presents the impact of prompt length and pseudo-label thresholds, adding detailed ablation studies on the contribution of different components would strengthen the claims about design choices.
2. While the use of a shared projection layer for cross-modal interaction is explained, the underlying mechanism for how this enhances feature alignment could be expanded, potentially with supporting theoretical or empirical analysis.
3. The paper could benefit from deeper analysis comparing its approach to more recent state-of-the-art methods.

**Questions:**

1. Could the authors provide a comprehensive ablation study to evaluate the impact of each component in the proposed method?
2. Can the authors elaborate on how the shared projection layer contributes to cross-modal feature alignment and provide supporting analysis?
3. How does the proposed method compare to recent state-of-the-art approaches such as [1]? Providing more detailed comparative results would add clarity.

[1] Du, Zhekai, et al. "Domain-agnostic mutual prompting for unsupervised domain adaptation." Proceedings of the IEEE/CVF Conference on Computer Vision and Pattern Recognition. 2024.

**Details Of Ethics Concerns:**

None.

---

### Official Review · Reviewer_vzSX · 2024-11-03

**Soundness:** 3
**Presentation:** 2
**Contribution:** 2
**Rating:** 3
**Confidence:** 4

**Summary:**

This paper propose a finetuning framework that integrates prompts with multimodal Low-Rank Adaptation (LoRA) for Multi-Source Domain Adaptation.

**Strengths:**

This paper propose a finetuning framework that integrates prompts with multimodal Low-Rank Adaptation (LoRA) for Multi-Source Domain Adaptation. They considers a class-specific prompt for MSDA and introduces a shared projection layer to aggregate features from different modalities when designing lora.

**Weaknesses:**

The presentation is poor. For example, x denotes the input image in line 173 while x turns into the visual representation of input image. In line 53, the citations should be placed before the period. In line 261, the 'bb learnable tokens'; in line 265, '[CLASS][CLASS] is replaced according to the label'. Equation 6 does not show the class-specific prompting as the authors stated. In equation 8, had should be replace with head.
The novelty seems limited.
The experiments are limited. For example, the ablation study is absent and the baseline methods are limited.

**Questions:**

see above weakness.

---

### Official Review · Reviewer_pCmr · 2024-11-03

**Soundness:** 2
**Presentation:** 2
**Contribution:** 2
**Rating:** 3
**Confidence:** 4

**Summary:**

This paper explores the issue of MUDA, which is a crucial and practical research direction. It introduces a fine-tuning framework that combines prompts with multi-modal Low-Rank Adaptation (LoRA). This framework utilizes trainable prompts as common elements across various domains and leverages multi-modal LoRA matrices to capture domain-specific features. Throughout the training process, these multi-modal LoRA matrices interact with a shared mapping matrix. The experimental results are emphasized.

**Strengths:**

1. The problem of MUDA is practical and worth solving.
2. The experiment results are good.

**Weaknesses:**

1. The novelty and contribution of this work are quite constrained. The interaction mechanism is borrowed from the MMA paper [1], thus it cannot be deemed as a unique contribution.
2. The figure is not a vector graphic, so it appears blurry.
3. In Eq. (6), this form does not represent a class-specific prompt, as the trainable component is shared across all classes.
4. Figure 1 is plagiarized from Figure 2 of the MMA paper [1].
5. Mistake in Line 285: '$\gamma$is a scaling factor' $\rightarrow$ '$\gamma$ is a scaling factor' .
6. Mistake In Line 288, What does LL mean? L?
7. Mistake in Line 253: 'However, Li et al. Li et al. (2023)'.
8. Mistake in Line 260: 'We introduce bb learn-able tokens'
9. Mistake in Line 265: 'Where, [Class] $[CLASS]$ is replaced according to the label'.
10. Mistake in Line 302: what does "L" refer to on Earth? Does it denote the text encoder "L" or the "L-th" transformer block?
11. Mistake in Line 303: "modify the equation1as".
12. Mistake In equation.(8), what is had? head？
13. The author mentioned applying low-rank matrices to the attention matrices within the transformer structure, but this approach seems to deviate from the formulations in Eq. (11) and Eq. (13). Further, these equations bear a resemblance to those presented in the MMA paper.
14.  In Eq. (11) and Eq. (13), the class-specific prompts are not included, where are they?
15. In Fig.(2), there are just too many errors and unexplained parameters.
16. The experiments are not sufficient, at least you should compare your methods with MMA [1].


[1]Lingxiao Yang, Ru-Yuan Zhang, Yanchen Wang, and Xiaohua Xie. Mma: Multi-modal adapter for vision-language models. In Proceedings of the IEEE/CVF Conference on Computer Vision and Pattern Recognition (CVPR), pp. 23826–23837, June 2024a.

**Questions:**

​See above.

---

### Note · Authors · 2024-11-17

I have read and agree with the venue's withdrawal policy on behalf of myself and my co-authors.